# Decoder Only Transformer for Physics-Informed Neural Networks

## Abstract

Physics-Informed Neural Networks (PINNs) approximate PDE solutions by embedding physical constraints into training, yet MLP-based backbones often suffer from instability and loss of fidelity on long horizons. Recent sequence models (e.g., Transformers) alleviate some of these issues, but their encoder–decoder design adds parameters and memory pressure with limited benefit for autoregressive pseudo-sequences. We introduce **DoPformer**, a *decoder-only* Transformer tailored to physics-informed learning. DoPformer consumes short spatio–temporal pseudo-sequences, uses multi-head self-attention with WaveAct activations, and applies a sequential physics loss across the window. Removing the encoder and cross-attention yields a lighter model while preserving long-range temporal coupling through self-attention. To further boost spectral accuracy, we explore two optional modules: (i) a Fourier *neural-operator* branch (*DoPformer+NO*) that improves oscillatory regimes and long-horizon rollouts; and (ii) a compact *KAN*-based feed-forward replacement (*DoPformer+KAN*) that drastically reduces parameters while maintaining strong accuracy. Across convection, reaction, wave, and 2D Navier–Stokes equations, DoPformer consistently improves PINN accuracy and stability; the NO and KAN variants deliver additional gains depending on stiffness and spectral content. Our numerical results show that on these benchmarks DoPformer attains state-of-the-art accuracy among physics-informed models while using substantially fewer parameters.

## 1 Introduction

Numerically solving partial differential equations (PDEs) has long been a central problem in science and engineering. Classical numerical solvers, such as the finite element method (Bathe, 2008) or pseudo-spectral method (Fornberg, 1996), provide accurate solutions but incur high computational cost, particularly in high-dimensional or multiscale settings. With the rise of scientific machine learning, Physics-Informed Neural Networks (PINNs) (Lagaris et al., 1997; Raissi et al., 2019) have emerged as a promising alternative. PINNs approximate the solution $u_\theta(x, t)$ using a neural network trained by minimizing a physics-informed objective that combines PDE residuals with initial and boundary conditions. This mesh-free, data-free paradigm has been successfully applied to many forward and inverse PDE problems.

Despite their flexibility, conventional PINNs built on multilayer perceptrons (MLPs) often fail when solutions involve oscillatory, high-frequency, or multiscale components (Raissi & Karniadakis, 2018; Fuks & Tchelepi, 2020; Krishnapriyan et al., 2021; Wang et al., 2022). Such models tend to produce over-smoothed solutions that satisfy residuals locally but fail to propagate information from initial conditions globally. These failure modes are especially pronounced for hyperbolic PDEs (e.g., convection, wave), where accurate temporal coupling is critical.

Two broad routes have been explored to mitigate these issues. One leverages additional data or sampling strategies (Raissi et al., 2017; Zhu et al., 2019; Faroughi et al., 2023), which may be impractical in data-scarce regimes. The other modifies optimization and training schemes (Krishnapriyan et al., 2021), often at substantial computational cost. A complementary approach is to strengthen architectural inductive biases. Recent work adapts sequence models to PINNs: Transformer-based PINNs (Zhao et al., 2024) capture temporal dependencies via encoder–decoder attention, while state-

space models (e.g., PINN-Mamba) (Xu et al., 2025) align subsequences to combat over-smoothing. These ideas improve accuracy but introduce additional complexity and parameters.

**Why decoder-only?**  We revisit the design of sequence models for PINNs through the lens of the training signal and data geometry. In physics-informed learning, inputs and targets inhabit the *same* spatio–temporal manifold: each token is a coordinate $(x, t)$, and the model predicts $u(x, t)$ at that token. Unlike supervised sequence-to-sequence settings that map between heterogeneous domains (e.g., translation), there is no distinct source/target stream that would necessitate cross-attention. Thus, encoder layers and cross-attention can be redundant, yet they add parameters, memory traffic, and extra Jacobian–vector products for automatic differentiation through the residual. By analogy with modern language modeling where decoder-only Transformers excel at autoregressive inference on homogeneous token streams, a decoder-style self-attention stack over short pseudo-sequences should be sufficient (and preferable) for PINNs: self-attention propagates temporal information across the window, while removing the encoder reduces optimization stiffness, activation memory, and FLOPs without sacrificing capacity.

**Our work.**  We propose **DoPformer**, a streamlined *decoder-only* Transformer tailored for physics-informed PDE solving. DoPformer consumes short spatio–temporal pseudo-sequences, uses multi-head self-attention with WaveAct activations, and applies a sequential physics loss across the window. The architecture is simple yet effective: by removing the encoder and cross-attention, it achieves higher accuracy with significantly fewer parameters. To further improve spectral fidelity, we augment the backbone with a **Fourier operator** branch (DoPformer+NO) that captures high-frequency modes and stabilizes long-horizon rollouts, and we explore a compact **KAN**-based feed-forward replacement (DoPformer+KAN) that injects spline/symbolic inductive bias with only a few thousand parameters.

**Contributions.**

- We introduce **DoPformer**, a decoder-only Transformer for PINNs that avoids encoder–decoder redundancy and delivers stronger accuracy with fewer parameters.

- We develop **spectral augmentation** via a lightweight Fourier (neural-operator) branch, improving oscillatory/high-frequency regimes and long-horizon stability.

- We propose a **KAN** feed-forward variant that achieves extreme parameter efficiency ($\sim 3K$ params) while maintaining high accuracy on smooth problems.

- Through comprehensive experiments on reaction, convection, wave, and 2D Navier–Stokes, we show that DoPformer matches or surpasses strong sequence baselines (including PINN-Mamba) while being the most lightweight competitive model; ablations quantify the complementary roles of the streamlined backbone, Fourier augmentation, and KAN feed-forward.

In summary, DoPformer establishes a simple, accurate, and efficient recipe for physics-informed PDE learning: a decoder-style attention backbone for temporal coupling, optional spectral augmentation for high-frequency content, and a compact feed-forward alternative for parameter-critical regimes.

## 2 PRELIMINARIES

**Physics–informed learning.**  Let $u : \Omega \times [0, T] \to \mathbb{R}^{d_{\text{out}}}$ be the solution of a PDE with operator $\mathcal{N}$, boundary operator $\mathcal{B}$, and initial data $u_0$:

$$\mathcal{N}[u](\boldsymbol{x}, t) = \boldsymbol{0}, \qquad (\boldsymbol{x}, t) \in \Omega \times (0, T], \qquad (1)$$

$$\mathcal{B}[u](\boldsymbol{x}, t) = \boldsymbol{0}, \qquad (\boldsymbol{x}, t) \in \partial\Omega \times [0, T], \qquad (2)$$

$$u(\boldsymbol{x}, 0) = u_0(\boldsymbol{x}), \qquad \boldsymbol{x} \in \Omega. \qquad (3)$$

PINNs parameterize $u$ by a neural network $u_{\boldsymbol{\theta}}(\boldsymbol{x}, t)$ and optimize a physics loss that penalizes residuals of equation 1–equation 3. With distributions over interior, boundary, and initial points,

$\mathcal{D}_{\Omega \times [0,T]}$, $\mathcal{D}_{\partial\Omega \times [0,T]}$, and $\mathcal{D}_{\Omega}$, the population objective is

$$\mathcal{L}(\boldsymbol{\theta}) = \lambda_r \, \mathbb{E}_{(\boldsymbol{x},t) \sim \mathcal{D}_{\Omega \times [0,T]}} \left[ \left\| \mathcal{N}[u_{\boldsymbol{\theta}}](\boldsymbol{x},t) \right\|_2^2 \right] \tag{4}$$

$$+ \lambda_b \, \mathbb{E}_{(\boldsymbol{x},t) \sim \mathcal{D}_{\partial\Omega \times [0,T]}} \left[ \left\| \mathcal{B}[u_{\boldsymbol{\theta}}](\boldsymbol{x},t) \right\|_2^2 \right] \tag{5}$$

$$+ \lambda_0 \, \mathbb{E}_{\boldsymbol{x} \sim \mathcal{D}_{\Omega}} \left[ \left\| u_{\boldsymbol{\theta}}(\boldsymbol{x},0) - u_0(\boldsymbol{x}) \right\|_2^2 \right]. \tag{6}$$

In practice we use Monte Carlo estimates on finite sets $\mathcal{D}_r = \{(\boldsymbol{x}_i, t_i)\}_{i=1}^{N_r}$, $\mathcal{D}_b = \{(\boldsymbol{x}_j, t_j)\}_{j=1}^{N_b}$, $\mathcal{D}_0 = \{\boldsymbol{x}_\ell\}_{\ell=1}^{N_0}$:

$$\widehat{\mathcal{L}}(\boldsymbol{\theta}) = \lambda_r \, \frac{1}{N_r} \sum_{i=1}^{N_r} \left\| \mathcal{N}[u_{\boldsymbol{\theta}}](\boldsymbol{x}_i, t_i) \right\|_2^2 \tag{7}$$

$$+ \lambda_b \, \frac{1}{N_b} \sum_{j=1}^{N_b} \left\| \mathcal{B}[u_{\boldsymbol{\theta}}](\boldsymbol{x}_j, t_j) \right\|_2^2 \tag{8}$$

$$+ \lambda_0 \, \frac{1}{N_0} \sum_{\ell=1}^{N_0} \left\| u_{\boldsymbol{\theta}}(\boldsymbol{x}_\ell, 0) - u_0(\boldsymbol{x}_\ell) \right\|_2^2. \tag{9}$$

All derivatives in $\mathcal{N}$ and $\mathcal{B}$ (e.g., $\partial_t u_{\boldsymbol{\theta}}$, $\nabla_{\boldsymbol{x}} u_{\boldsymbol{\theta}}$, $\Delta u_{\boldsymbol{\theta}}$) are obtained by automatic differentiation. We follow the common PINN convention of non-dimensionalizing variables and using fixed weights $\lambda_r, \lambda_b, \lambda_0$ unless noted; task-specific operators and domains are given in Sec. 4.

**Limitations of MLP-based PINNs.** Most PINNs employ point-wise multilayer perceptrons (MLPs) that map $(\boldsymbol{x}, t) \mapsto u(\boldsymbol{x}, t)$ independently across coordinates. Despite universal approximation, such models frequently underperform on oscillatory or multiscale PDEs, yielding over-smoothed solutions that appear to satisfy residuals at sampled collocation points but fail globally Krishnapriyan et al. (2021); Fuks & Tchelepi (2020). Two factors recur in analyses: (*i*) lack of temporal inductive bias—point-wise predictors do not explicitly propagate information from initial conditions across time, which is critical for transport- and wave-dominated regimes; (*ii*) optimization bias toward simple hypotheses—training can settle on overly smooth or trivial patterns that minimize discrete residuals yet violate the true dynamics between samples Wang et al. (2022). These limitations motivate sequence-aware PINNs that encode temporal coupling within each training window.

**Pseudo-sequences and Transformers (PINNsFormer).** To inject temporal inductive bias, PINNsFormer maps a single query $(\boldsymbol{x}, t)$ to a short *pseudo-sequence*

$$\mathcal{S}_k(\boldsymbol{x}, t; \Delta t) = \{[\boldsymbol{x}, t], [\boldsymbol{x}, t + \Delta t], \dots, [\boldsymbol{x}, t + (k-1)\Delta t]\} \in \mathbb{R}^{k \times d_{\text{model}}},$$

where $[\cdot]$ concatenates spatial and temporal coordinates, $k$ is the window length, $\Delta t$ is the stride, and $d_{\text{model}}$ is the embedding width Zhang et al. (2024). Each token in $\mathcal{S}_k$ is linearly embedded ("spatio–temporal mixer"), then the window is processed by a Transformer with an *encoder–decoder* stack. The encoder applies self-attention and feed-forward layers; the decoder, unlike the vanilla Transformer, *omits decoder self-attention* and keeps only encoder–decoder attention plus an FFN, reusing the same embedded tokens as queries/keys/values (no separate target sequence). A small output head predicts the field for all tokens. Inside FFNs, PINNsFormer uses a learnable wavelet-style activation $\omega_1 \sin(\cdot) + \omega_2 \cos(\cdot)$ to enhance spectral expressivity Zhang et al. (2024).

**Sequential physics loss.** The objective is switched from point-wise to sequence-wise: residual and boundary terms are averaged across the $k$ tokens, while the initial-condition penalty is applied only to the first token (the earliest time in the window),

$$\mathcal{L}_{\text{seq}} = \lambda_r \, \frac{1}{k} \sum_{j=0}^{k-1} \mathbb{E} \left[ \left\| \mathcal{N}[u_{\theta}](\boldsymbol{x}, t + j\Delta t) \right\|_2^2 \right] \tag{10}$$

$$+ \lambda_b \, \frac{1}{k} \sum_{j=0}^{k-1} \mathbb{E} \left[ \left\| \mathcal{B}[u_{\theta}](\boldsymbol{x}, t + j\Delta t) \right\|_2^2 \right] \tag{11}$$

$$+ \lambda_0 \, \mathbb{E} \left[ \left\| u_{\theta}(\boldsymbol{x}, t) - u_0(\boldsymbol{x}) \right\|_2^2 \right]. \tag{12}$$

This encourages temporal propagation of constraints within each window and empirically improves generalization on convection, reaction, and wave problems Zhang et al. (2024).

**Kolmogorov–Arnold networks (KAN).** KANs Liu et al. (2025) replace fixed-node activations in MLPs by *learnable univariate edge functions* with linear aggregation at nodes, motivated by the Kolmogorov–Arnold representation. For a layer with $n_\ell$ inputs and $n_{\ell+1}$ outputs, collect edge functions in $\Phi^{(\ell)} = \{\phi_{j,i}^{(\ell)}\}_{j=1,i=1}^{n_{\ell+1},\,n_\ell}$ and define

$$z_j^{(\ell+1)} = \sum_{i=1}^{n_\ell} \phi_{j,i}^{(\ell)}(z_i^{(\ell)}), \qquad j = 1, \dots, n_{\ell+1}, \tag{13}$$

so a depth-$L$ KAN is $f(x) = (\Phi^{(L-1)} \circ \cdots \circ \Phi^{(0)})(x)$. Each edge function is parameterized as a residual–spline

$$\phi(x) = w_b\, b(x) + w_s \sum_{r=1}^{G+k} c_r\, B_r^{(k)}(x), \tag{14}$$

where $b(x)$ is a fixed base nonlinearity (e.g., SiLU), $\{B_r^{(k)}\}$ are B–spline bases of order $k$ on a grid with $G$ intervals, and $\{w_b, w_s, \{c_r\}\}$ are trainable coefficients. This construction yields compact, spectrally expressive univariate transforms on each edge while retaining simple additive aggregation at nodes. All operations are differentiable, hence compatible with automatic differentiation required by the physics residuals. Empirically, the learnable edge activations provide rich local function classes (e.g., sinusoidal/decay profiles) that are frequently encountered in PDE solutions, making KAN a convenient drop-in alternative to standard pointwise activations in PINN backbones .

**Fourier neural operators (FNO).** Neural operators learn maps between *functions* rather than fixed-size vectors. A Fourier neural operator layer updates a feature field by (i) transforming to the Fourier domain, (ii) applying learnable complex multipliers on the lowest modes, and (iii) transforming back:

$$\widehat{v}'(k) = R_\phi(k)\,\widehat{v}(k) \text{ for } |k| \le k_{\max}, \qquad v^+ = \sigma\big(\mathcal{F}^{-1}(\widehat{v}') + Wv\big), \tag{15}$$

where $\widehat{v} = \mathcal{F}(v)$ is the FFT of the field along the relevant axis (spatial or the short pseudo-sequence), $R_\phi(k)$ are learnable spectral weights on a truncated band of modes, $W$ is a pointwise linear map, and $\sigma$ is a nonlinearity. This provides global mixing via a few Fourier modes and local refinement via $W$, yielding strong accuracy on oscillatory or multiscale PDEs and good resolution transfer Li et al. (2021); Kovachki et al. (2021).

## 3 DoPformer

### 3.1 Design motivation

Sequence-aware PINNs (Sec. 2) suggest that *temporal coupling inside a short window* is the key inductive bias to prevent over-smoothing and to propagate initial conditions. However, the encoder–decoder layout of PINNsFormer duplicates computation on the same token window and pays extra for cross-attention Zhang et al. (2024); Vaswani et al. (2017). **DoPformer** removes this redundancy: we keep only *self-attention over the pseudo-sequence* and a strong pointwise nonlinearity, and we add optional spectral modules that target the remaining failure modes (high-frequency drift). This leads to higher accuracy per parameter and simpler optimization, while preserving the sequential physics loss from Sec. 2.

### 3.2 Architecture

**Overview.** Given a batch of pseudo-sequences $\{\mathcal{S}_k(\boldsymbol{x}, t; \Delta t)\}$ (Sec. 2), we concatenate spatial and temporal scalars per token and apply a linear embedding to obtain $X^{(0)} \in \mathbb{R}^{B \times k \times d_{\text{model}}}$. **DoPformer** stacks $N$ *decoder-only* Transformer blocks that operate along the $k$-token window: each block performs multi-head self-attention (MHSA) over tokens followed by a position-wise feed-forward network (FFN); both sublayers use pre-activation WaveAct. A lightweight head maps tokens to field values. There is no encoder and no cross-attention.

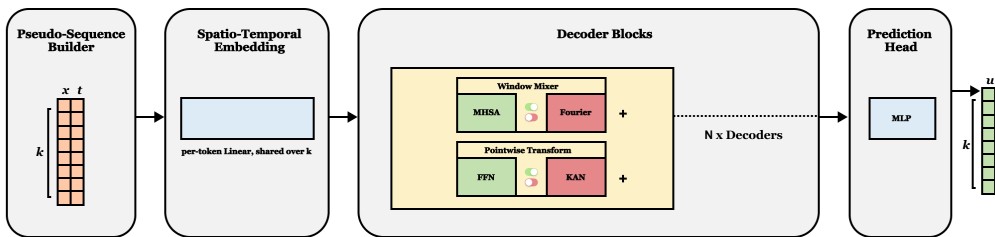

Figure 1: **DoPformer overview.** The *Pseudo-Sequence Builder* expands a single query $(x, t)$ into a $k$-token window $\mathcal{S}_k(x, t; \Delta t)$. The *Spatio-Temporal Embedding* applies a shared per-token linear map $\mathbb{R}^2 \to \mathbb{R}^{d_{\text{model}}}$. A stack of $N$ *Decoder Blocks* then processes this representation. In each block, the top section **Window Mixer** mixes along the $k$-token window using **either** multi-head self-attention (**MHSA**) **or** a spectral **Fourier** operator. The bottom section **Pointwise Transform** applies per-token **either** a standard **FFN** (Linear → WaveAct → Linear) **or** a compact **KAN** (two KAN layers with B-spline bases and LayerNorm). Both sections use *pre-activation WaveAct* and *residual* connections. The *Prediction Head* is a small MLP with WaveAct that projects $B \times k \times d_{\text{model}}$ to $B \times k \times d_{\text{out}}$; at inference we read the token corresponding to the query time. Configurations: **BASE** = MHSA + FFN; **+NO** = Fourier + FFN; **+KAN** = MHSA + KAN; **+NO+KAN** = Fourier + KAN.

**Windowing and tokens.** A token is the concatenation $[\boldsymbol{x}, t]$. For a query $(\boldsymbol{x}, t)$ we form a short window $\mathcal{S}_k(\boldsymbol{x}, t; \Delta t) = \{[\boldsymbol{x}, t + i\Delta t]\}_{i=0}^{k-1}$. Batches comprise many such windows sampled over $\Omega \times [0, T]$. We do not add extra positional encodings: the absolute time $t + i\Delta t$ inside each token and the short window length are sufficient in practice.

**Wavelet activation.** We adopt the wavelet-style nonlinearity Zhang et al. (2024)

$$\text{WaveAct}(z) = \boldsymbol{\omega}_1 \odot \sin(z) + \boldsymbol{\omega}_2 \odot \cos(z), \tag{16}$$

with trainable $\boldsymbol{\omega}_1, \boldsymbol{\omega}_2$, which improves spectral expressivity and stabilizes multiscale training.

**Core decoder-only block.** Let $X \in \mathbb{R}^{B \times k \times d_{\text{model}}}$ be the block input. One **DoPformer** block updates $X$ as

$$X \leftarrow X + \text{MHSA}\big(\text{WaveAct}(X)\big), \tag{17}$$

$$X \leftarrow X + \text{FFN}\big(\text{WaveAct}(X)\big). \tag{18}$$

We use no causal mask (the window is local in time) and no cross-attention. This decoder-only stack mixes information across the $k$ tokens and avoids the encoder–decoder duplication in Zhang et al. (2024).

**Neural-Operator block (exact spec).** In **DoPformer+NO** we *replace* MHSA by a spectral operator acting along the $k$-token window:

$$X \leftarrow X + \text{NO}\big(\text{WaveAct}(X)\big), \qquad X \leftarrow X + \text{FFN}\big(\text{WaveAct}(X)\big).$$

We use no normalization and no gating. The implementation treats the window as a (latitude $= k$, longitude $= 1$) grid, applies spectral mixing, and projects back to $[B, k, D]$. **Hyperparameters (used in our experiments):** embedding $D{=}32$, heads$= 2$, window $k{=}7$ (Reaction), bias enabled; fusion is by *replacement* (NO instead of MHSA), no concatenation. For other PDEs we keep the same recipe and only adjust $k$ with the pseudo-sequence schedule.

**KAN feed-forward (exact spec).** In **DoPformer+KAN** we *replace* FFN by a two-layer KAN block with pre-activation WaveAct and a residual connection, without normalization. Each edge uses a quadratic B-spline basis with $G{=}6$ intervals ($G{+}k{=}8$ bases) plus a SiLU base:

$$\phi(x) = w_b \, \text{SiLU}(x) + w_s \sum_{r=1}^{8} c_r \, B_r^{(2)}(x), \quad \Phi(z)_j = \sum_{i=1}^{n_{\text{in}}} \phi_{j,i}(z_i).$$

KAN-FFN stacks two such layers $\Phi^{(1)} : \mathbb{R}^{d_{\text{model}}} \to \mathbb{R}^{\gamma d_{\text{model}}}$ and $\Phi^{(2)} : \mathbb{R}^{\gamma d_{\text{model}}} \to \mathbb{R}^{d_{\text{model}}}$, with $d_{\text{model}}{=}8$, $\gamma{=}2$ (hidden${=}16$) in the parameter-efficient Reaction setup, yielding $\approx 3.16$k parameters for the whole model. In **DoPformer+NO+KAN** both replacements are applied (NO for MHSA, KAN for FFN).

**Block update**

$$Y \leftarrow X + \text{NO/MHSA}\big(\text{WaveAct}(X)\big)$$
$$Z \leftarrow Y + \text{FFN/KAN}\big(\text{WaveAct}(Y)\big)$$
**return** $Z$

```
// Internals of NO (shape transforms)
q4 = query.permute(0,2,1).unsqueeze(-1)          % [B,D,k,1]
o4 = attention(q4, k4, v4)                       % spectral mixing
out = o4.squeeze(-1).permute(0,2,1)              % back to [B,k,D]
```

**Head and outputs.** A small MLP with WaveAct projects $X \in \mathbb{R}^{B \times k \times d_{\text{model}}}$ to $\hat{U} \in \mathbb{R}^{B \times k \times d_{\text{out}}}$, producing predictions at all tokens. During training we use the sequential physics loss (Sec. 2); at inference for a single $(\boldsymbol{x}^*, t^*)$ we form $\mathcal{S}_k(\boldsymbol{x}^*, t^*; \Delta t)$ and read the prediction at the corresponding token.

### 3.3 SEQUENTIAL PHYSICS LOSS

We train with the sequence objective from Sec. 2: residual and boundary terms are averaged across the $k$ tokens of each window, and the initial condition is enforced only at the earliest token Zhang et al. (2024). This couples local temporal neighborhoods and propagates constraints without requiring an encoder or cross-attention.

## 4 EXPERIMENTS

**Training details.** For each PDE we sample interior, boundary, and initial collocation sets as in Sec. 2, resampling interior points every fixed number of iterations. All models are trained with L-BFGS to convergence under the same stopping rule; learning-rate and line-search settings are shared. For sequence models we use the sequential loss. The pseudo-sequence hyperparameters $(k, \Delta t)$ are aligned with prior work Zhang et al. (2024); Xu et al. (2025) on each benchmark. Inputs $(\boldsymbol{x}, t)$ are non-dimensionalized and standardized per task. Unless noted, we do not use dropout or causal masks; gradients are clipped only on divergence.

**Configurations (our methods).** We evaluate four DoPformer variants: (i) **DoPformer**: decoder-only Transformer with MHSA+FFN and WaveAct; (ii) **DoPformer+NO**: same backbone but replacing MHSA with a Fourier neural-operator layer acting along the $k$-token window (spectral attention); (iii) **DoPformer+KAN**: FFN replaced by a compact KAN block per token (two KAN layers with cubic B-splines on a coarse grid); (iv) **DoPformer+NO+KAN**: combination of spectral attention and KAN FFN. Model widths/depths/heads are chosen to match the parameter ranges reported in the result tables.

**Benchmarks (equations and settings).** We assess our methods on the standord set of benchmark equations, namely:

*Convection (1D advection).*

$$\partial_t u + \beta \, \partial_x u = 0, \qquad x \in [0, 2\pi], \; t \in [0, 1],$$

with periodic boundaries and $u(x, 0) = \sin x$; larger $\beta$ increases transport dominance.

*Reaction (1D logistic).*

$$\partial_t u - \rho \, u(1 - u) = 0, \qquad x \in [0, 2\pi], \; t \in [0, 1],$$

with periodic boundaries and a localized initial bump $u(x, 0)$; stiffness grows with $\rho$.

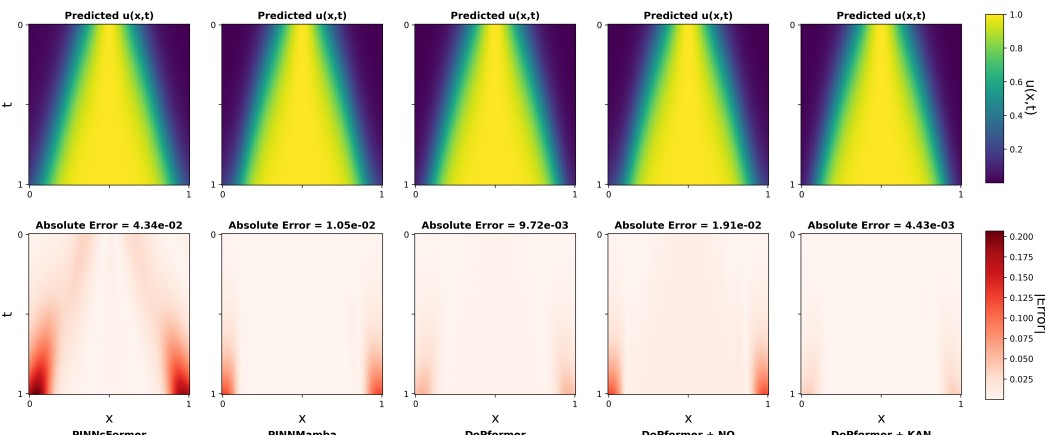

Figure 2: **Qualitative comparison on a 1D spatio–temporal benchmark.** *Top:* predicted fields $u(x,t)$. *Bottom:* pointwise absolute-error maps $|\hat{u} - u|$; the value above each panel is the mean absolute error over the space–time grid. Columns: **PINNsFormer**, **PINNMamba**, **DoPformer**, **DoPformer + NO** (Fourier neural-operator mixing along the short pseudo-sequence), and **DoP-former + KAN** (compact KAN feed-forward head). The decoder-only backbone reduces diffusion artifacts; the spectral branch (**+NO**) recovers high-frequency content, while the KAN variant attains the lowest pointwise error in this example. Experimental settings are in Sec. 4; quantitative results are in Tabs. 1

*Wave (1D).*
$$\partial_{tt}u - c^2 \, \partial_{xx}u = 0, \qquad x \in [0,1], \ t \in [0,1],$$

$$u(0,t) = u(1,t) = 0, \qquad u(x,0) = \sin(\pi x) + \tfrac{1}{2}\sin(\beta\pi x), \qquad \partial_t u(x,0) = 0,$$

stressing spectral fidelity due to multi-frequency superposition.

*Navier–Stokes (2D, incompressible).*
$$\partial_t \mathbf{v} + (\mathbf{v}\cdot\nabla)\mathbf{v} = -\nabla p + \nu \, \Delta\mathbf{v}, \qquad \nabla\cdot\mathbf{v} = 0, \qquad (x,y) \in \Omega, \ t \in [0,1],$$

with standard no-slip/inflow boundary conditions; we report errors for $(u, v, p)$ on dense test grids under the same setup as sequence-PINN baselines.

**Metrics.** We report total physics loss at convergence and relative errors on dense test grids of the original resolution: $\mathrm{rMAE} = \frac{\sum_n |\hat{u}_n - u_n|}{\sum_n |u_n|}$ and $\mathrm{rRMSE} = \sqrt{\frac{\sum_n \|\hat{u}_n - u_n\|_2^2}{\sum_n \|u_n\|_2^2}}$. We also list trainable parameter counts to assess efficiency.

**Results: Convection & Reaction.** On *convection*, **DoPformer** achieves the lowest rMAE/rRMSE among all methods while using markedly fewer parameters than sequence baselines. This supports the hypothesis that, for transport-dominated regimes, decoder-only self-attention over short windows suffices to propagate ICs without encoder–decoder overhead. On *reaction*, **DoPformer+KAN** yields the best errors overall, indicating that stronger token-wise nonlinearity (via KAN) is advantageous when dynamics are locally stiff but less oscillatory; the spectral branch (**+NO**) is robust yet not essential here.

**Results: Wave & 2D Navier–Stokes.** For the *wave* equation, attention-only **DoPformer** under-fits high-frequency content as expected; adding the Fourier branch (**DoPformer+NO**) substantially reduces error, validating spectral augmentation along the window. On *2D Navier–Stokes*, the combined **DoPformer+NO+KAN** variant attains the strongest accuracy among our models while remaining extremely compact, demonstrating that low-rank spectral mixing and expressive token non-linearities complement each other in multi-field, higher-dimensional flows.

Table 1: Results for solving **convection** and **reaction** equations.

| Model | #Params | Convection | | | Reaction | | |
|---|---|---|---|---|---|---|---|
| | | Loss | rMAE | rRMSE | Loss | rMAE | rRMSE |
| PINN | 527361 | 0.0239 | 0.8514 | 0.8989 | 0.1991 | 0.9803 | 0.9785 |
| QRes | 396545 | 0.0798 | 0.9035 | 0.9245 | 0.1991 | 0.9826 | 0.9830 |
| PINNsFormer | 453561 | 0.0068 | 0.4527 | 0.5217 | 3e-6 | 0.0434 | 0.0686 |
| KAN | 891 | 0.0250 | 0.6049 | 0.6587 | 7e-6 | 0.0166 | 0.0343 |
| PINN-Mamba | 285763 | 4.1e-5 | 0.0188 | 0.0201 | 1e-6 | 0.0105 | 0.0248 |
| DoPformer | 161295 | 0.0001 | **0.0145** | **0.0165** | 2e-6 | 0.0097 | 0.0169 |
| DoPformer+NO | 161295 | 0.0002 | 0.0243 | 0.0381 | 3e-6 | 0.0191 | 0.0209 |
| DoPformer+KAN | 3159 | 0.0001 | 0.0796 | 0.0932 | **1e-6** | **0.0043** | **0.0090** |
| DoPformer+NO+KAN | 3159 | 0.0001 | 0.0436 | 0.0932 | 9.8e-6 | 0.0564 | 0.0746 |

Table 2: Results for solving **wave** and **2D Navier–Stokes** equations.

| Model | #Params | Wave | | | Navier–Stokes (2D) | | |
|---|---|---|---|---|---|---|---|
| | | Loss | rMAE | rRMSE | Loss | rMAE | rRMSE |
| PINN | 527361 | 0.0320 | 0.4101 | 0.4141 | 7.31e-5 | 14.42 | 10.02 |
| QRes | 396545 | 0.0987 | 0.5349 | 0.5265 | 2.24e-4 | 6.41 | 4.45 |
| PINNsFormer | 453561 | 0.0216 | 0.3559 | 0.3632 | 6.49e-6 | 0.375 | 0.274 |
| KAN | 891 | 0.0067 | 0.1433 | 0.1458 | 3.43e-4 | 8.74 | 7.02 |
| PINN-Mamba | 285763 | 0.0002 | 0.0197 | 0.0199 | 1.26e-5 | 2.128 | 1.074 |
| DoPformer | 161295 | 0.0002 | **0.0173** | **0.0178** | 5.63e-6 | 0.278 | 0.222 |
| DoPformer+NO | 161295 | 0.0002 | 0.0201 | 0.0207 | 5.53e-6 | 0.285 | 0.213 |
| DoPformer+KAN | 3159 | 0.0003 | 0.0351 | 0.0407 | 3.11e-5 | 2.453 | 1.642 |
| DoPformer+NO+KAN | 3159 | 0.0002 | 0.0202 | 0.0211 | 3.76e-6 | **0.176** | **0.104** |

**Efficiency.** The decoder-only design eliminates encoder self-attention and cross-attention, cutting parameters and FLOPs versus encoder–decoder Transformers and lowering activation memory (beneficial for LBFGS). In our settings, **DoPformer** uses ∼40% fewer weights than PINN-Mamba and ∼3× fewer than PINNsFormer, yet matches or surpasses their accuracy. The NO branch adds only a small spectral module (FFT and a few complex weights) and an optional gating projection; the KAN swap keeps the FFN budget tiny (few thousand parameters) while boosting per-token expressivity.

**Discussion.** KAN brings the most benefit on reaction-type dynamics: its learnable univariate edge functions provide strong token-wise nonlinearity that captures sharp local responses and helps with stiffness, without adding sequence-mixing complexity. For wave-like problems, a Fourier neural-operator layer along the short window supplies the missing spectral mixing, reducing the low-frequency bias of attention-only backbones and stabilizing long-horizon rollouts. Because pseudo-sequences are formed around a single spatio–temporal query, encoder and decoder in an encoder–decoder stack effectively process the same tokens; a decoder-only design removes duplicated projections yet keeps the crucial inductive bias of temporal coupling within the window. Finally, the compact DoPformer+NO+KAN configuration scales well to 2D multi-field systems, offering a practical route to higher-dimensional PDEs without the complexity typically associated with heavy sequence encoders.

## 5 RELATED WORK

PINNs solve PDEs by enforcing physics residuals at collocation points Raissi et al. (2019), yet training can fail on multiscale/oscillatory regimes or exhibit imbalance among loss terms Krishnapriyan et al. (2021). Remedies include domain decomposition (XPINNs) for scalability and discontinuities Jagtap & Karniadakis (2020) and causal curricula to stabilize long-horizon transients Wang et al. (2024). Sequence-aware models introduce temporal inductive bias: PINNsFormer builds short pseudo-sequences with Transformer attention and spectral activations Zhang et al. (2024), while PINN-Mamba leverages state-space sub-sequences to mitigate continuous–discrete mismatch and simplicity bias Xu et al. (2025). Compact backbones reduce parameters and sharpen local re-

sponse, e.g., KAN-based designs such as KINN and AL-PKAN Wang et al. (2025); Zhang et al. (2025). Frequency-aware approaches curb spectral bias by operating in the Fourier domain or enriching high-frequency bases Yu et al. (2024). Finally, physics-informed operator learning (e.g., FNO/PINO) targets generalization across PDE families via spectral convolutions with residual-based constraints Li et al. (2021; 2024).

## 6 CONCLUSION

We introduced **DoPformer**, a streamlined decoder-only Transformer for physics-informed learning of PDEs. By coupling short pseudo-sequences with lightweight multi-head self-attention and WaveAct-enhanced token updates, DoPformer retains the temporal inductive bias needed for transport- and wave-dominated dynamics while discarding the encoder–decoder redundancy of prior sequence PINNs. We further showed that two orthogonal augmentations—(i) a Fourier neural-operator layer for spectral mixing along the window and (ii) a compact KAN-based feed-forward module for expressive token-wise nonlinearities—can be plugged in without disrupting the physics loss or automatic differentiation.

Across four standard benchmarks (1D convection, reaction, wave; 2D Navier–Stokes), DoPformer variants match or surpass strong baselines, including recent state-of-the-art sequence models, while using far fewer parameters. Ablations indicate clear regimes of benefit: KAN excels on locally stiff reaction dynamics; the Fourier branch is essential for oscillatory wave problems; and their combination scales favorably to multi-field 2D flows. Together, these results support a simple message: for pseudo-sequence PINNs, decoder-only temporal mixing is sufficient and often preferable when paired with targeted spectral and nonlinearity enhancements.

**Limitations and future work.** Our study focuses on compact windows and fixed $(k, \Delta t)$ schedules; adaptive windowing, causal masks for extrapolative rollouts, and multi-resolution spectral mixing are natural extensions. Scaling to 3D and multi-physics systems, incorporating hard boundary constraints and geometry encoders, and integrating uncertainty quantification or data assimilation are promising directions. Finally, unifying neural-operator layers with KAN-style token nonlinearities inside a single block may further improve accuracy–efficiency trade-offs for challenging PDE regimes.

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
