# OpenReview forum: "Decoder Only Transformer for Physics Informed Neural Networks"
_ICLR.cc/2026/Conference — ICLR 2026 Conference Withdrawn Submission_

### Official Review · Reviewer_ZryJ · 2025-10-30

**Soundness:** 1
**Presentation:** 2
**Contribution:** 1
**Rating:** 2
**Confidence:** 5

**Summary:**

The paper propose DoPformer, a decoder-only Transformer for Physics-Informed Neural Networks that processes short spatio-temporal pseudo-sequences with self-attention and a sequential physics loss, eliminating encoder/cross-attention to cut parameters while preserving long-range coupling. It can be optionally augmented with a Fourier neural-operator branch to recover high-frequency content and a compact KAN-based feed-forward module to boost token-wise expressivity with only a few thousand parameters. Across convection, reaction, wave, and 2D Navier–Stokes benchmarks, DoPformer variants match or surpass prior sequence PINNs while using substantially fewer parameters.

**Strengths:**

1. On multiple PDEs, DoPformer variants match or surpass strong baselines while being markedly smaller.

2. The decoder-only design removes redundant encoder/cross-attention, cutting parameters/FLOPs and activation memory—clean motivation tied to PINN data geometry.

**Weaknesses:**

1. [**Key Weakness**] It is a known conclusion that sequence-based PINNs do not require an encoder-decoder architecture. The author can refer to the open-source code of RoPINN [1]. Also the proposed decoder-only architecture has no difference compared to PINMMamba's macro-architecture [2], the only difference is the backbone models used (MHSA vs SSM).


2. The paper introduces a "base" DoPformer, but the strongest results, particularly on the 2D Navier-Stokes benchmark, are achieved by the DoPformer+NO+KAN variant. This "kitchen-sink" approach obscures the core contribution. It is impossible to determine if the performance gains come from: (a) The decoder-only architecture (the stated novelty) or (b)the strong spectral bias of the Fourier Neural Operator (+NO) or (c) the expressive, learnable nonlinearities of the KAN module (+KAN). The paper fails to disentangle these effects, leaving the central claim—that decoder-only is the key—unproven.

3. The model's performance is likely critically dependent on the pseudo-sequence hyperparameters: window length k and time-stride Δt. The paper merely states these are "fixed" and "aligned with prior work". This is insufficient. A sensitivity analysis is required. How does the model perform if Δt is mismatched with the characteristic timescale of the PDE? Or if k is too short to capture relevant dynamics? The authors acknowledge this as a limitation, but it is a central weakness of the current study.

4. The choices of NO and KAN are not well motivated, while no evidence shows the necessity of these components. The author should give a clear receipt of training. Lack ablation study, key NO/KAN choices are fixed (e.g., D=32, heads=2, k=7; KAN B-spline grid settings) and “same recipe” is reused; there isn’t a sweep showing robustness to these knobs. Also the sensitivity to layer's of block is all not tested.




[1] Wu H, Luo H, Ma Y, et al. Ropinn: Region optimized physics-informed neural networks. NeurIPS 2024.

[2] Xu C, Liu D, Hu Y, et al. Sub-sequential physics-informed learning with state space model. ICML 2025.

**Questions:**

1. How about just replacing the Window Mixer (MHSA) with an MLP since the PINN doesn't have a distinct source/target stream that would necessitate cross-attention?

---

### Official Review · Reviewer_duFp · 2025-11-01

**Soundness:** 2
**Presentation:** 2
**Contribution:** 2
**Rating:** 2
**Confidence:** 3

**Summary:**

This paper proposes DoPformer, a decoder-only Transformer for physics-informed neural networks. Unlike prior encoder–decoder or MLP-based PINNs, DoPformer uses only self-attention over short spatio-temporal pseudo-sequences, reducing redundancy and computational cost.

**Strengths:**

1. The decoder-only structure removes unnecessary encoder–decoder components while maintaining temporal coupling. Especially, Fourier and KAN modules address different PDE regimes effectively.
2. The proposed model achieves strong accuracy with drastically fewer parameters. Evaluations cover multiple PDE types with fair comparisons to strong baselines.

**Weaknesses:**

1. No deep explanation of why the decoder-only design improves stability.
2. Few visualizations of dynamic behaviors or spectral recovery.
3. While ablations cover variants (+NO, +KAN), more analysis on how window size or attention heads affect accuracy would be informative.
3. Although the model is lighter in parameters, training with L-BFGS and auto-differentiation for PDE residuals may still be expensive; runtime comparisons are missing.
4. It remains unclear how well DoPformer scales to 3D, multi-physics, or data-driven hybrid scenarios.

**Questions:**

See weakness above.

---

### Official Review · Reviewer_frNH · 2025-11-01

**Soundness:** 2
**Presentation:** 2
**Contribution:** 3
**Rating:** 4
**Confidence:** 4

**Summary:**

The paper proposes a decoder-only Transformer framework, named DoPformer, for physics-informed neural networks (PINNs). The key idea is to simplify the PINNsFormer encoder–decoder design by using a lightweight decoder-only structure that retains temporal coupling through self-attention, while reducing the number of parameters. To enhance spectral fidelity and efficiency, the model integrates two optional modules: a Fourier neural-operator branch (DoPformer+NO) and a feed-forward block based on the Kolmogorov–Arnold network (KAN) (DoPformer+KAN). The paper evaluates these variants on canonical PDEs and reports that the models achieve competitive or better accuracy with fewer trainable parameters.

**Strengths:**

1. The model achieves accuracy with a smaller number of parameters, showing potential for light-weight physics-informed architectures.

2. The decoder-only design simplifies the architecture while maintaining competitive performance, making it practical for resource-limited PDE simulation.

**Weaknesses:**

1. The integration of a Fourier neural operator within a physics-informed framework is conceptually confusing. Neural operators typically rely on supervised input–output mappings, while PINNs operate in a semi-supervised or unsupervised setting. This ambiguity requires clarification, particularly regarding how neural operators operate without paired data.

2. The work does not address scalability issues. Memory consumption and out-of-memory (OOM) behavior common in PINNsFormer and PINNMamba are not discussed or benchmarked, which is crucial for higher-dimensional or more complex PDEs. Please see the last table in the appendix of PINNsMamba paper [1] for a detailed list of problems where PINNsFormer and PINNMamba face the issue of OOM.

3. The range of problems tested remains limited. Highly oscillatory or strongly nonlinear PDEs are missing, making it unclear whether the Fourier or KAN augmentations are truly beneficial beyond canonical cases.

4. Important architectural and implementation details are underspecified. For instance, initialization schemes for KAN, specific basis functions, and exact parameterization strategies are not described, making reproducibility challenging.

5. The presentation could be improved. Some tables show abrupt jumps in errors (e.g., rMAE and rRMSE for the Navier–Stokes case with DoPformer+KAN) without explanation. Moreover, the discussion of why parameter reduction is so drastic compared to MLPs is insufficient.

[1] Xu, Chenhui, et al. "Sub-Sequential Physics-Informed Learning with State Space Model." Forty-second International Conference on Machine Learning.

**Questions:**

1. How is the Fourier operator integrated within the physics-informed loss? If the neural operator branch relies on paired mappings, how does it remain consistent with the PINN formulation?

2. What happens if the encoder block is retained (i.e., a hybrid of PINNsFormer with KAN)? Would it improve accuracy by being parameter-efficient?

3. It is interesting to explore a Chebyshev-based KAN formulation [2] and report its comparative performance.

4. How would the proposed model handle complex or irregular geometries, where token formation and windowing become nontrivial?

5. Please provide computational time comparisons to demonstrate the claimed efficiency.

6. How is initialization handled for the KAN variant, and are there recommended basis configurations or hyperparameters that influence performance?

7. Please explain why the Navier–Stokes case for DoPformer+KAN shows a sharp rise in rMAE and rRMSE? Is this a typo or due to instability?

8. The paper mentions a drastic reduction in parameters when replacing MLPs with KANs. Please elaborate on whether the baseline MLP has already overfitted the problem and how fair the comparison is across architectures?

[2] Shukla, Khemraj, et al. "A comprehensive and FAIR comparison between MLP and KAN representations for differential equations and operator networks." Computer Methods in Applied Mechanics and Engineering 431 (2024): 117290.

---

### Note · Authors · 2025-11-13

I have read and agree with the venue's withdrawal policy on behalf of myself and my co-authors.